# Nomogram for Early Prediction of Outcome in Coma Patients with Severe Traumatic Brain Injury Receiving Right Median Nerve Electrical Stimulation Treatment

**DOI:** 10.3390/jcm11247529

**Published:** 2022-12-19

**Authors:** Chao Zhang, Wen-Dong You, Xu-Xu Xu, Qian Zhou, Xiao-Feng Yang

**Affiliations:** 1Emergency and Trauma Center, The First Affiliated Hospital, Zhejiang University School of Medicine, Hangzhou 310003, China; 2Department of Neurosurgery, Minhang Hospital, Fudan University School of Medicine, Shanghai 201100, China

**Keywords:** coma, traumatic brain injury, RMNS, EEG, nomogram, prognosis

## Abstract

Background: Accurate outcome prediction can serve to approach, quantify and categorize severe traumatic brain injury (TBI) coma patients for right median electrical stimulation (RMNS) treatment, which can support rehabilitation plans. As a proof of concept for individual risk prediction, we created a novel nomogram model combining amplitude-integrated electroencephalography (AEEG) and clinically relevant parameters. Methods: This study retrospective collected and analyzed a total of 228 coma patients after severe TBI in two medical centers. According to the extended Glasgow Outcome Scale (GOSE), patients were divided into a good outcome (GOSE 3–8) or a poor outcome (GOSE 1–2) group. Their clinical and biochemical indicators, together with EEG features, were explored retrospectively. The risk factors connected to the outcome of coma patients receiving RMNS treatment were identified using Cox proportional hazards regression. The discriminative capability and calibration of the model to forecast outcome were assessed by C statistics, calibration plots, and Kaplan-Meier curves on a personalized nomogram forecasting model. Results: The study included 228 patients who received RMNS treatment for long-term coma after a severe TBI. The median age was 40 years, and 57.8% (132 of 228) of the patients were male. 67.0% (77 of 115) of coma patients in the high-risk group experienced a poor outcome after one year and the comparative data merely was 30.1% (34 of 113) in low-risk group patients. The following variables were integrated into the forecasting of outcome using the backward stepwise selection of Akaike information criterion: age, Glasgow Coma Scale (GCS) at admission, EEG reactivity (normal, absence, or the stimulus-induced rhythmic, periodic, or ictal discharges (SIRPIDs)), and AEEG background pattern (A mode, B mode, or C mode). The C statistics revealed that the nomograms’ discriminative potential and calibration demonstrated good predictive ability (0.71). Conclusion: Our findings show that the nomogram model using AEEG parameters has the potential to predict outcomes in severe TBI coma patients receiving RMNS treatment. The model could classify patients into prognostic groups and worked well in internal validation.

## 1. Introduction

Traumatic brain injury (TBI) is a severe life-threatening illness and has become a big social challenge [1,2]. Numerous TBI patients become comatose. Coma is an unconscious state characterized by the failure of the brain’s arousal and alerting system (ascending reticular activation system, ARAS) [3]. Coma patients who suffered from severe TBI cannot always be woken with conventional medical interventions [4]. Some patients can regain consciousness within the first few days after severe trauma, but some enter a protracted coma or vegetative state, which is a form of wakeful unawareness [5]. Others progress into brain death [6]. The best outcome that the family of patients and physical therapists desired is that patients can be effectively treated to restore recovery during the acute period.

Right median electrical stimulation (RMNS) is an important consciousness-promoted rehabilitation therapy used to hasten awakening from coma [6]. It took more than two decades to establish the right median nerve electrical stimulation (RMNS) application in coma awakening. Researchers from the United States and Japan indicated RMNS can accelerate the consciousness of coma patients [7,8]. Four representative scientific clinical studies verified the capacity of RMNS treatment to arouse coma patients [7,9,10,11] but 40% of post-traumatic coma patients still suffered unsatisfactory neurological outcome after RMNS treatment [11]. However, the assessment of coma patients’ prognosis after RMNS is difficult. This calls for multi-dimensional clinical information of the patient for consideration. The details cover information of demographics, etiology, disease severity, laboratory testing, imaging, and electroencephalogram (EEG). Such multi-dimensional research on predicting the course of coma patients treated with RMNS therapy is, however, lacking at the moment.

Amplitude-integrated electroencephalography (AEEG) is frequently a component of multimodal monitoring on TBI patients treated in neurological intensive care units (N-ICU), where it is mostly used to identify (non-convulsive or electrographic) seizures [12,13]. Various initiatives have been made for correlating EEG characteristics to the neurological outcome of TBI patients [13,14,15,16]. As its background pattern feature and normal EEG reactivity have shown good evaluation value in predicting brain injury [17,18,19], AEEG has become a convenient and potential neurological outcome indicator for individuals in a coma after TBI. To screen out the ideal types of patients who could benefit from the RMNS treatment, we proposed a novel nomogram with the amplitude-integrated electroencephalography (AEEG) application to build one early prediction model.

## 2. Methods

### 2.1. Study Participants

From November 2016 to December 2021, consecutive coma patients who were at the two-week stage following severe TBI in our study centers routinely received RMNS as consciousness-promoted rehabilitation therapy. A total of 228 patients were finally retrospectively collected and analyzed from the intensive care unit (ICU) of The First Affiliated Hospital, Zhejiang University School of Medicine, and the neurosurgical intensive care unit (N-ICU) in Renji Hospital, Shanghai Jiaotong University School of Medicine. The inclusion criteria were as follows: (1) the age of 18 years or older; (2) a confirmed history of TBI caused by a traffic accident, a fall, or other causes, and intracranial contusion and hematomas, subarachnoid hemorrhage, or diffuse axonal injury could be diagnosed with a CT scan; (3) a GCS score ≤8 on admission, with the GCS score continuing to be under or equal to 8 following one week of intense therapy; (4) prolonged coma (>7 days); (5) a stable condition with respect to the vital signs and the intracranial imaging on CT scan, for example no diffuse brain swelling or active bleeding. The exclusion criteria were as follows: (1) severe heart arrhythmias or pacemaker placement in the past; (2) epilepsy; (3) pregnancy; (4) decreased life expectancy and co-occurring systemic disorders.

### 2.2. Prognostic Assessment

Each recruited patient was followed up on by one or more investigators at the clinic or by videophone 12 months following discharge. The purpose of our research was to explore the characteristics of ideal patient for receiving RMNS consciousness–promoted treatment, and whether awakening is our primary concern. The primary outcome measurement was the extended Glasgow Outcome Scale (GOSE), which was divided into poor outcome (GOSE 1–2) or good outcome (GOSE 3–8). One year later is the time node that we mainly focus on. 

### 2.3. Clinical Parameters 

Age, gender, pupillary responses, and laboratory parameters (hemoglobin and arterial partial pressure of oxygen, PO_2_) at admission were recorded. According to the electronic health record system, the following data were retrospectively collected at the beginning of RMNS treatment: GCS and mean arterial blood pressure (MAP). Age is a significant predicting factor in TBI recovery [20]. and hypotension is related to poor outcome and increased mortality after TBI [21,22,23].

### 2.4. RMNS Programing

The recruited comatose patients received RMNS after one-week standard neurosurgical therapy according to the guidelines [24,25]. To conduct the electrical therapy, a couple of lubricated, 1-inch-square rubberized surface electrodes were used. They were adhered to the right distal forearm’s volar side above the medulla. An electrical neuromuscular stimulator was utilized by Verity Medical Ltd. in the UK to deliver trains of asymmetric biphasic pluses with amplitudes of 10–20 milliamps and pulse lengths of 300 s at 40 Hz lasting 20 s/min. The duration of the RMNS treatment was one week, at 10 hours per day [8].

### 2.5. AEEG Monitoring and Analysis

Before patients were treated with RMNS, AEEG was used to perform 3-day-long synchronous monitoring for exclusion of epilepsy, affirmation of patients’ elementary AEEG background patterns and other AEEG related indicators (EEG reactivity, absence, or presence of sleep-related waveforms). Two AEEG analysis experts in our study center, blinded to patients’ identity and clinical condition, examined the traces of AEEG, respectively. The raw EEG data were checked to certify the AEEG evaluation.

According to the modified criterion of Hellstrom-Westas et al. and Toet et al., the AEEG background patterns were categorized into five subtypes [26,27,28].

(1)Flat tracing (FT): continuous low-voltage pattern with an upper margin of <5 μV;(2)Continuous extremely low voltage (CLV): continuous low-voltage pattern with an upper margin of <10 μV and a lower margin of <5 μV;(3)Burst suppression (BS): discontinuous pattern, with periods of very low voltage intermixed with high amplitude and a lower margin constantly at 0–1 μV and a burst amplitude of >25 μV more than 50% of the recording time;(4)Discontinuous normal voltage (DNV): electrical attenuation (with an upper margin of >10 μV) or suppression occurring (a lower margin of <5 μV) 10–49% of the recording time.(5)Continuous normal voltage (CNV): continuous pattern with a low voltage margin of 5–10 μV and an upper voltage margin of 10–50 μV. Only with sporadic electrical attenuation or suppression (<10% of the record).

Most CNV cases appeared in full-term asphyxiated neonates who returned to normal or shallow sleep within 24 hours. BS, CLV, and FT cases were related to the cortical electrical activity of full-term asphyxiated neonates with poor prognosis, while cases of DNV were identified in the AEEG of both normal and pathological neonates [28,29]. Referring to the classification of previous studies [17,30,31], all of the patients with a CNV background patterns were divided into A mode, those with DNV background patterns were divided into B mode, and others with BS, CLV, and FT background patterns were divided into C mode in our research.

EEG reactivity was defined as alteration in the EEG generated by stimulus. When patients were given moderate external stimulations such as painful (pinching the nipple or limbs), visual (opening eyes under light), and auditory (continuing to call out the patient’s name) [32,33]. If the patients’ EEGs displayed reactivity changes in frequency or amplitude, this demonstrated they had EEG reactivity; otherwise, there was no EEG reactivity. Reactivity was not defined as electromyographic activity or blink artifacts. SIRPIDs (stimulus-induced rhythmic, periodic, or ictal discharges) are abnormal discharges caused primarily by vigilant stimulation such as sensory stimulus, chest constriction, limb constraint, turning, and other nursing intervention.

Vertex sharp waves, spindles, and K-complexes were identified as sleep-related waveforms because they were clearly recognizable and related to the sleep phase. [34,35]. We defined the presence of sleep-related waveforms as the appearance of one or more waveforms throughout monitoring.

It was inevitable for patients to use central nervous system (CNS) drugs (propofol, chlorpromazine, diazepam, etc.) in the intensive care unit, and these neurological drugs may affect the results of AEEG. According to the rate of metabolism of the drugs in the blood, we excluded the period (15–30 mins) of applying neurological drugs when analyzing the AEEG data. This criterion was also applied when using drugs (dexamethasone, indomethacin) and temperature maintenance equipment to correct hyperpyrexia or hypothermia.

### 2.6. Statistical Analysis

Unless otherwise indicated, continuous variables were presented as medians with interquartile ranges (IQRs), while categorical variables were given as round numbers and proportions. The Kaplan-Meier method was used to produce the outcome for the research population. A priori, clinical and EEG variables associated with outcome were evaluated according to the clinical significance, scientific cognition, and predicting variables identified in previously research [19,36,37]. A correlation matrix was applied to test explanatory variables for co-linearity, as well as plausible interaction subjects, such as interactions between age, GCS at admission, EEG reactivity (normal, absence, or SIRPIDs), and AEEG background pattern, were investigated (A mode, B mode, or C mode). Since no significant correlation was discovered, no interaction component was contained at the multivariable analysis. To be comparable with previous data [17,19], the AEEG background pattern was modeled as a categorical variable (A mode, B mode, or C mode), and EEG reactivity as a categorical variable (absence, SIRPIDs vs normal). Cox proportional hazards regression models were performed to assess the relationships between pertinent clinical and EEG variables and outcome. The variables for the multiple Cox proportional hazards regression models were identified using backward stepwise selection with the Akaike information criterion (AIC). Using statistical software, some characteristics were incorporated into the nomograms to forecast the likelihood of coma patients’ 1-year favorable outcomes following RMNS treatment (rms in R, version 3.6.3). We use regression coefficients to construct the linear predictor for each individual observation when assigning points in the nomograms.

The efficiency of the nomograms was assessed by C statistics. The C statistic, which is equal to the area under the receiver operating characteristic curve, estimates the likelihood of accordance between predicted and observed outcomes [38]. Calibration was tested with the bootstrapped sample of the research cohort and a calibration plot. Predictions in a well-calibrated model would drop on the 45-degree diagonal line. To further evaluate calibration, we constructed Kaplan-Meier curves over the dichotomy of coma patients stratified by the points predicted by the nomograms. Data was then analyzed statistically using software programs (SPSS version 29.0 and R, version 3.6.3), and *p* < 0.05 was considered statistically significant.

## 3. Results

### 3.1. Demographic and Clinical Physiology Characteristics

A total of 228 patients at two centers were finally included. The median age of patients was 40 years (IQR, 33.0–47.0 years), and 57.9% (132 of 228) of the patients were male (Table 1). On admission, 32.0% (73 of 228) of patients had a bilateral reacting pupillary response, 39.9% (91 of 228) with unilateral reacting pupillary response, and 28.1% (64 of 228) with no reacting pupillary response on both sides. The median GCS score was 5 (IQR, 4–6). A minority of the patients had hyphemia (36.0% [82 of 228]), or hypoxia (29.4% [67 of 228]). Emergency surgery was performed on 63.6% (145 of 228) of the patients. The AEEG was dynamically monitored for 3-day period from the fourth day after the patient’s admission. On the final AEEG report, most patients (61.4% [67 of 228]) showed B mode in AEEG background pattern, with the remaining patients exhibiting C mode (11.4% [26 of 228]) and A mode (27.2% [62 of 228]), respectively. Among all of the patients, most had a normal EEG reactivity (47.8% [109 of 228]) or SIRPIDs (37.3% [85 of 228]), while only 14.9% (34 of 228) of the patients presented absence in EEG reactivity. The sleep-related wave can be detected in 55.7% (127 of 228) of the patients. In our study, a patient with one or several kinds of complications after TBI was defined as the existence of a complication, including neurological complications: hydrocephalus, cerebrospinal fluid (CSF) leakage, intracranial infection, and non-neurological complications: bedsore, pneumonia, liver and kidney insufficiency. Before receiving RMNS therapy, most patients had an appropriate MAP (76.3% [174 of 228]), and no complications (86.0% [196 of 228]). 

The median follow-up time was 12.7 months (range, 4.3–38.2 months); 26 patients died (GOSE 1) and 58 stayed in a persistent vegetative state (GOSE 2) within 1 year. Five patients were missed for follow-up because they unable to be contacted via telephone. Then, based on medical records, their outcomes were evaluated; three of them were in a vegetative state upon discharge and allocated to the poor outcome group (GOSE 1–2), whereas the other two patients were allocated to the good outcome group (GOSE 3–8). In total, 111 of 228 (48.7%) patients had a poor outcome at 1 year, and the remaining 117 (51.3%) patients had a good outcome. A flow diagram for including and excluding eligible coma patients is represented in Figure 1.

### 3.2. Model Configuration and Predictors of the Outcome

Established risk factors, together with demographic and clinical physiological characteristics, were chosen as candidate variables for the prognosis model. In the Cox proportional hazards regression modeling, the backward stepwise selection revealed the four variables with the strongest association with a favorable outcome (GOSE 3–8): age, GCS at admission, EEG reactivity, and AEEG background pattern (Table 2). On multivariable analysis, age (HR, 1.02; 95% CI, 1.00–1.04; *p* = 0.013), GCS (HR, 0.83; 95% CI, 0.68–1.02; *p* = 0.032), SIRPIDs (HR, 1.76; 95% CI, 1.16–2.68; *p* = 0.008), and A (HR, 0.45; 95% CI, 0.27–0.73; *p* < 0.001), B (HR, 0.29; 95% CI, 0.16–0.50; *p* < 0.001) mode of AEEG background pattern were each independently related to a good outcome.

### 3.3. Nomograms Performance

The nomograms predicting a good outcome of severe TBI coma patients at 1 year are presented in Figure 2. The following four independent prognostic factors built the final nomogram model: age, GCS at admission, EEG reactivity (normal, absence, or SIRPIDs), and AEEG background pattern (A mode, B mode, or C mode). Higher total scores based on the nomograms were linked to a worse prognosis. As an example, a 40-year-old patient with GCS-5 at admission, evidence of normal EEG reactivity, and B mode in AEEG background pattern would recieve a total of 115 points (35 points for age, 45 points for GCS, 0 points for normal EEG reactivity, and 35 points for B mode), for a predicted 1-year good outcome of 61.0%. The model’s discriminative ability for outcome was also evaluated utilizing C statistics (0.71; 95% CI, 0.69–0.73).

To test the model’s discriminative capacity further, the predicted probability of a poor outcome was obtained by plotting as Kaplan-Meier curves stratified with the dichotomy of scores calculated from the total nomograms. The optimal cut-off value of the total nomogram scores was determined to be 140; then, patients with a total nomogram score above 140 were classified into high-risk group and those with low total nomogram scores into the low-risk group (Figure 3). Patients in the high-risk group had substantially higher 1-year and 2-year predicted poor outcomes (67.0% (77 of 115), and 45.2% (52 of 113), respectively) compared with patients in the low-risk group (30.1% (34 of 113), and 14.1% (16 of 113), respectively) (*p* < 0.001). Bootstrap validation with 200 re-samplings was used to assess model accuracy and potential model overfit. A 40-sample bootstrapped calibration plot for the forecast of a good outcome is shown in Figure 4.

## 4. Discussion

As an effective intervention treatment for coma arousal, RMNS was first proposed in 1999 [7] and proved that it can lead to faster emergence from a coma by several randomized trials [7,9,11]. Jin Lei et al., showed that the proportion of coma patients who regained consciousness in the RMNS group is comparatively higher than in the control group (59.8% vs. 46.2%, *p* < 0.01) [11]. While RMNS treatment remains the important consciousness-promoted choice for coma patients after severe TBI in our medical centers, approximately forty percent of the coma patients, who eventually stepped into death or vegetative state within one year, still cannot benefit from it. Therefore, we need to design a multifactor prediction model to precisely forecast the prognosis of coma patients who received RMNS, and further select patients suitable for this treatment. Furthermore, optimized outcome predictions may aid in the credible quantification and classification of coma and TBI severity [39].

The International Mission for Prognosis and Clinical Trial Design (IMPACT) predictor [36] and Corticosteroid Randomization after Significant Head Injury (CRASH) [37] predictor are currently the best influential predictors for neurological outcome after TBI. However, these prediction models, which are determined by parameters at admission, have not taken into account the progression of secondary injury in TBI patients, nor the influence of therapy [40]. Given the heterogeneity of the primary injury and the evolution of secondary injury in these patients, combining multiple AEEG features may reveal more factors related to the prognosis of coma TBI patients. We eventually developed a nomogram depending on patient-related and AEEG-related parameters that digitally predicted an individual’s outcome after RMNS treatment. A mode of AEEG background pattern, normal EEG reactivity, a low age, and a high GCS were the independent relevant factors for a good outcome in the outcome prognosis model with high accuracy.

Rundgren et al., first applied the AEEG pattern to predict prognosis in adult coma patients with cardiac arrest. It was shown that the continuous AEEG pattern (A mode) was closely associated with a higher survival rate and improved consciousness recovery. While an FT, BS (belonged to C mode), or status epilepticus AEEG was strongly predictive of an unfavorable outcome [41,42]. The prognostic value of the AEEG pattern for cardiac arrest patients who received hypothermia treatment was then affirmed by Oh et al. and Sugiyama et al. [43,44,45] Wendong You et al., demonstrated that the AEEG pattern is also a promising predictor of outcome for coma patients of varied etiologies including TBI and stroke [17]. In contrast to the previous studies, we found that coma patients in B mode had a nearly good prognosis as patients in A mode after receiving RMNS, with hazard ratios (HR) of 0.56 and 0.39 for patients in C mode, respectively (Table 2). This may be because coma patients have a more stable brain electrical activity basis in both A and B modes.

The brainstem reticular formation and the thalamocortical cortical loop are the anatomical foundations of EEG reactivity [46]. Some specific diseases, including severe TBI and hypoxic ischemic encephalopathy (HIE), can damage these structures, with the basal ganglia and neocortex of man being especially vulnerable, so that the absence of EEG reactivity signifies that these structures have been injured, with potential neurological implications [47]. The Synek and Young grading scales, which are both now recognized as EEG prognostic criteria, have both included the EEG reactivity as significant parameters [48,49]. According to prior research on the predicting outcome of coma patients, 48–92% of patients with EEG reactivity regain coma arousal after 5 months [49], while patients with no EEG reactivity have a mortality rate of up to 93% within 1.5 years [50]. SIRPIDs are a kind of abnormal EEG reactivity that can be induced by alertness stimuli in different etiologies [12]. Alvarez et al. monitored 114 cardiac arrest patients with continuous EEG (CEEG) monitoring and discovered that when patients emerged SIRPIDs, their outcome at 3 months was unfavorable [51]. In our study, the HR for normal EEG reactivity and SIRPIDs to the absence of EEG reactivity are 0.54 and 1.58, respectively. The study’s findings were in accordance with those of prior studies, which suggests that aberrant or nonexistent EEG reactivity is a sign of impaired brain function and a significant risk factor for unfavorable outcomes.

Age and GCS were both important components of our final nomogram model and made an effective definition of the overall physiological conditions of coma patients at admission. This is in line with the fundamental prognostic model of TBI in IMPACT and CRASH. Less significant to our models were gender, pupillary response, hypoxia, arterial PO_2_, MAP, and sleep-related wave. Though several studies have indicated that sleep-related waves were independently associated predictor of a good outcome in patients with acute encephalopathy [35,52], this lacks sufficient relevance for prognosis in our study, which may be related to the heterogeneity of origin disease in coma patients or due to the absence of subgroup analysis of sleep waves.

Considering the prediction of patients could be heterogeneous, appropriate risk stratification of coma patients before RMNS treatment is meaningful. Instead of applying staging information from the IMPACT or CRASH gradation classification, which is generated from population-relied or massive cohort data, nomograms could provide a more individualized and visible way to make a prognosis. When stratified into dichotomy by the optimal cut-off of nomogram accumulated scores, the proposed nomograms could distinguish between groups of coma patients who were at a high or low risk of a poor outcome (Figure 3). Moreover, our nomograms demonstrated a favorable discriminative ability, with a C statistic of 0.71 for forecasting a good outcome (Figure 2). When the nomogram model predicted 1-year survival, it resembled the actual survival as determined by Kaplan-Meier curves (Figure 4). Altogether, the data effectively suggests the proposed nomograms can provide individual patient information about likelihood of a certain outcome for coma patients after severe TBI receiving RMNS consciousness-promoted treatment.

Some studies have reported that RMNS therapy could provide neurological outcome benefits for coma patients with severe TBI [6,7,9,11]. While only about half of patients received RMNS, 40% of post-traumatic coma patients still could obtain satisfactory neurological outcomes after RMNS. In our study, 69.9% (79 of 113) of severe TBI coma patients in the low-risk group could have a good outcome at the 1-year follow-up. Personalized risk prediction models, such as the current nomograms, may be used to help choose ideal coma patients for RMNS and guiding rehabilitation treatment in the future. Although EEG measurements are time-consuming work, the prognostic model based on EEG parameters can not only diagnose epilepsy early but also provide timely intervention for coma patients suitable for RNMS treatment, and the benefits may outweigh the extra efforts. 

The current study had several limitations. As a retrospective study, there might be some unobserved or uncontrolled confounding factors and we might miss a few factors affecting the development of outcome among coma patients. Depending on the integrity of the measured data (such as clinical characteristic parameters, laboratory parameters, and AEEG data), the inability to control exposure and intervention of curing TBI are also the main limitations of this retrospective experiment. Therefore, the conclusion of this research only provides a reference of causality for large-scale prospective studies in the future. In addition, the GOSE does not take non-neurological causes of death into account (in the case of severe multiple trauma patients, pneumonia or bedsores caused by poor nursing and long-term bedridden eventually lead to death, and these patients in the low-risk group may receive a poor outcome) and that kind of patient’s survival mostly relies on clinical decision-making. Since awakening is our primary concern, we generally defined a good outcome as severe disability to good recovery (GOSE 3–8), while a good neurological recovery is also a concern for the clinician and patient’s family. The poor neurological outcome group defined as death to severe disability (GOSE 1–4) and good neurological outcome as the moderate disability to good recovery (GOSE 5–8) should also be further studied. Since we were unable to intervene in the treatment method, the possible effect of the CNS drugs application on the data derived from the AEEG measurement can only be corrected by excluding the EEG data at a brief period, which will still be biased depending on the accuracy of the AEEG parameters.

Another limitation of the study is that because the intent of this research was to utilize the data before RMNS treatment as an outcome predictor, it may fail to dynamically identify changes in the interval between RMNS. However, the principal purpose of this study is to find the target coma patients for electrotherapy early, and the early clinical characteristics prior to RMNS treatment are the main subjects of our concern. Certainly, large randomized clinical tests in the future to incorporate clinical and EEG data of changes before and after RMNS treatment can be conducted to further explore the prognostic impacts of this treatment on coma patients.

Finally, the findings lack generalizability, and more medical centers should collaborate in further study to externally validate the proposed nomograms.

## 5. Conclusions

The nomogram model, which incorporates AEEG traits and clinical data, has the capacity to forecast neurological outcome in coma patients after severe TBI receiving RMNS treatment.

## Figures and Tables

**Figure 1 jcm-11-07529-f001:**
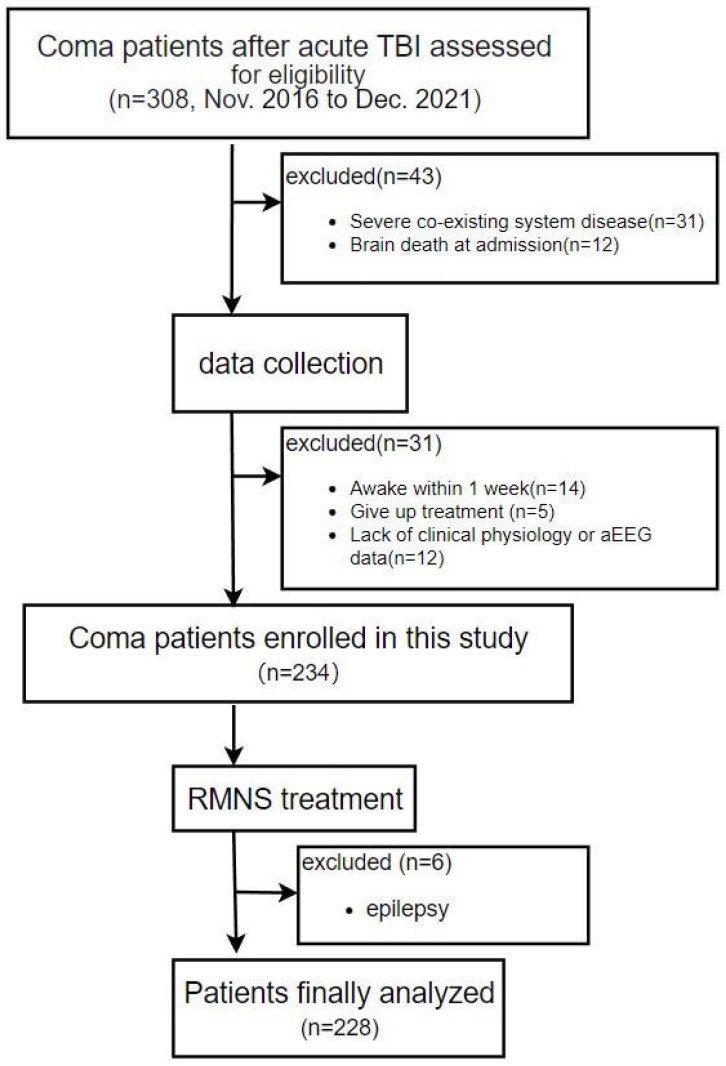
Recruitment flow chart of patients. Abbreviations: TBI, traumatic brain injury; RMNS, right median electrical stimulation.

**Figure 2 jcm-11-07529-f002:**
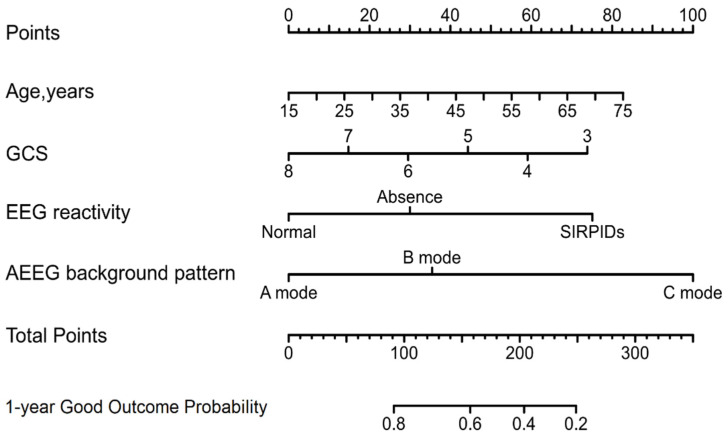
Nomogram Predicting Outcome in TBI Coma Patients After Receiving RMNS treatment. The nomogram to predict a good outcome was created based on four independent prognostic factors (see the Model Specifications and Predictors of good outcome subsection of the Methods section). Abbreviations: GCS, Glasgow Coma Scale; AEEG, amplitude-integrated electroencephalography.

**Figure 3 jcm-11-07529-f003:**
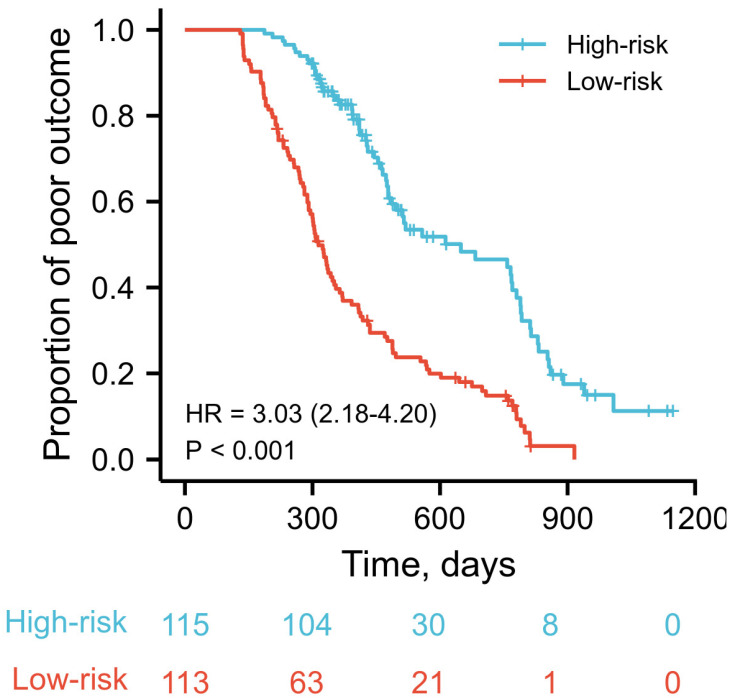
Kaplan-Meier Curves Demonstrating Poor Outcome in TBI Coma Patients After Receiving RMNS Treatment According to Dichotomy of Predicted Outcome. *p* values are by the log-rank test.

**Figure 4 jcm-11-07529-f004:**
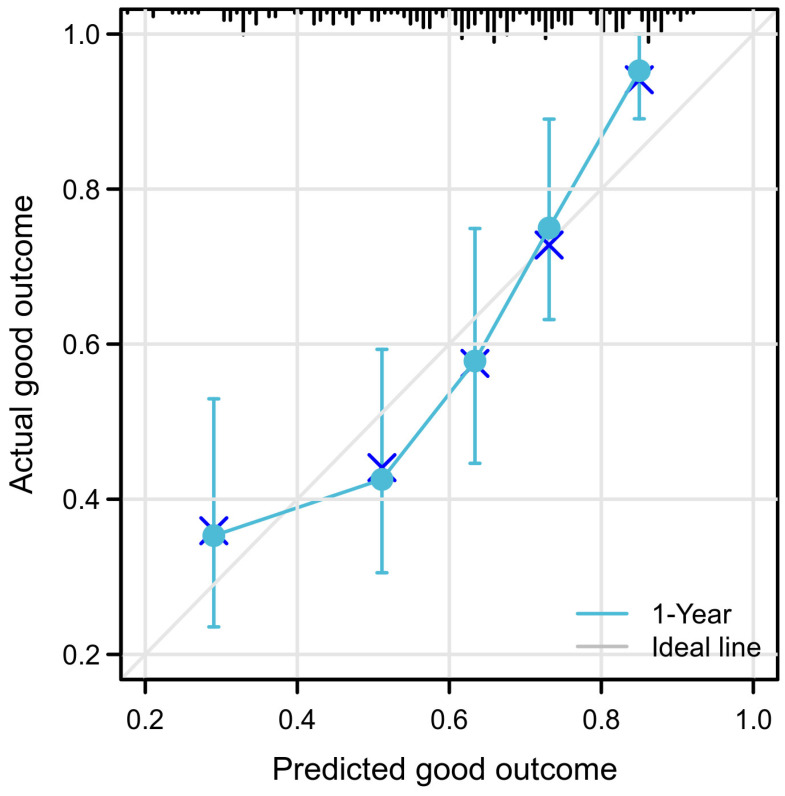
Calibration Plot Comparing Predicted and Actual Good Outcome Probabilities at 1-year Follow-up cross represents the result of each point corrected by hierarchical Kaplan-Meier’ after the word ‘bootstrap-corrected estimates’ in the annotation of Figure 1. The calibration plot for predicting a 1-year good outcome with 40 samples is presented. The gray line signifies best fit; circles show nomogram-predicted probabilities; crunode represents bootstrap-corrected estimates, and error bars stand for 95% confidence intervals for these estimates.

**Table 1 jcm-11-07529-t001:** Baseline Patient and AEEG Characteristics of the Patients Undergoing a Post-traumatic Coma.

Characteristic	Total (n = 228)
Age, years, median (IQR)	40.0 (33.0–47.0)
Gender, n (%)	
Male	132.0 (57.9)
Female	96.0 (42.1)
GCS, median (IQR)	5.0 (4.0–6.0)
Pupillary response, n (%)	
Both reacting	73.0 (32.0)
Single reacting	91.0 (39.9)
Neither reacting	64.0 (28.1)
Hyphemia, n (%)	
Yes	82.0 (36.0)
No	146.0 (64.0)
Arterial PO_2_, mmHg, n (%)	
<60	67.0 (29.4)
≥60	161.0 (70.6)
Surgical operation, n (%)	145.0 (63.6)
EEG reactivity, n (%)	
Absence	34.0 (14.9)
Normal	109.0 (47.8)
SIRPIDs	85.0 (37.3)
AEEG background pattern, n (%)	
C mode	26.0 (11.4)
B mode	140.0 (61.4)
A mode	62.0 (27.2)
Sleep-related wave, n (%)	
Absence	101.0 (44.3)
Presence	127.0 (55.7)
MAP, mmHg, n (%)	
<70	54.0 (23.7)
70–105	174.0 (76.3)
Complication, n (%)	
Existence	32.0 (14.0)
Absence	196 (86.0)

Abbreviations: MAP, mean arterial blood pressure; GCS, Glasgow Coma Scale; PO_2_, partial pressure of oxygen; AEEG, amplitude-integrated electroencephalography; SIRPIDs, the stimulus-induced rhythmic, periodic, or ictal discharges.

**Table 2 jcm-11-07529-t002:** Cox Proportional Hazards Regression Model Presenting the Association of Variables with a Good Outcome.

	Univariable		Multivariable	
Variable	HR (95% CI)	*p* Value	HR (95% CI)	*p* Value
**Factors Selected**				
Age, years	1.03 (1.02–1.05)	<0.001	1.02 (1.00–1.04)	0.013
GCS	0.68 (0.59–0.79)	<0.001	0.83 (0.68–1.02)	0.032
EEG reactivity				
Absence	1[Reference]	NA	1[Reference]	NA
Normal	0.54 (0.35–0.84)	0.006	0.69 (0.43–1.09)	0.011
SIRPIDs	1.58 (1.04–2.37)	0.030	1.76 (1.16–2.68)	0.008
AEEG background pattern				
C mode	1[Reference]	NA	1[Reference]	NA
B mode	0.56 (0.35–0.91)	0.020	0.29 (0.16–0.50)	<0.001
A mode	0.39 (0.18–0.54)	<0.001	0.45 (0.27–0.73)	<0.001
**Factors Not Selected**				
Gender				
Male	1[Reference]	NA	NA	NA
Female	0.93 (0.61–1.49)	0.54	NA	NA
Pupillary response				
Both reacting	1[Reference]	NA	NA	NA
Single reacting	2.96 (0.75–4.03)	0.41	NA	NA
Neither reacting	0.76 (0.38–0.93)	0.24	NA	NA
Hypoxia				
Yes	1[Reference]	NA	NA	NA
No	1.43 (0.98–2.64)	0.60	NA	NA
Arterial PO_2_, mmHg				
<60	1[Reference]	NA	NA	NA
≥60	1.67 (0.84–3.04)	0.31	NA	NA
MAP, mmHg				
<70	1[Reference]	NA	NA	NA
≥70	1.13 (0.76–2.13)	0.57	NA	NA
Sleep-related wave				
Absence	1[Reference]	NA	NA	NA
Presence	0.86 (0.44–1.24)	0.28	NA	NA

Abbreviations: NA, not applicable; HR, hazard ratio; GCS, Glasgow Coma Scale; AEEG, amplitude-integrated electroencephalography; PO_2_, partial pressure of oxygen; MAP, mean arterial blood pressure.

## Data Availability

The data presented in this study are available on request from the first author (C.Z.).

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
