# Peer review of "Nomogram for Early Prediction of Outcome in Coma Patients with Severe Traumatic Brain Injury Receiving Right Median Nerve Electrical Stimulation Treatment"

_jcm, 2022, doi:10.3390/jcm11247529_

Round 1

Reviewer 1 Report

This is an interesting paper dealing the prognosis of patients in come following TBI with severe trauma

The subject is addressed over the years by many groups as this is a very common problem faced by clinicians treating head trauma patients. The ability to predict the prognosis and outcome of the different head trauma patients is still being learned.

This study is trying to give a nomogram with a few parameters that help understand how bad is the injury and what are the chances of recovery and good function in the long term.

Age and GCS are well known factors. The EEG added in this nomogram, both background pattern and EEG reactivity to RMNS treatment are found to be corelated with the Neurological outcome. Not all Intensivists and Neurosurgeons treating these patients are familiar enough with the EEG. If it will prove to be an important predictor as shown in this paper, it will be added to our understanding and predicting the outcome of the severe TBI patients.

Author Response

Response: Thank you very much for giving us pertinent comments on our manuscript. We have carefully and repeatedly revised the language errors and simplified the manuscript with the necessary simplification(Please see the attachment). We hope that the revised version can meet the qualification.

Reviewer 2 Report

Dear Authors, I read your paper with interest. Prognosis after TBI or cardiac arrest is extremely important and difficult at the same time.

Your work could contribute to a better prognostication after TBI. Results are very interesting even if the retrospective design of the study poses some limits that should be acknowledged.

There are some English errors, so I encourage you to submit the paper for an English revision.

At the end of table 1, what do you mean with "Complication"?

Please, shorten as much as you can the discussion.

Best regards

Author Response

Point 1: There are some English errors, so I encourage you to submit the paper for an English revision.

Response 1: We have carefully and repeatedly revised the existing English errors in the manuscript, and hope that the revised version can meet the qualification.

Point 2: At the end of table 1, what do you mean with "Complication"?

Response 2: Patients with one or several complications before RMNS treatment was defined as the presence of complications, including neurological complication: hydrocephalus, cerebrospinal fluid (CSF ) leakage, intracranial infection, and non-neurological complications: bedsore, pneumonia, Liver and kidney insufficiency. We have revised and supplemented the instructions in both the form and in the text. Besides, there were too many types of complications to include them all, we did not list them as a study variable in Table 2.

Point 3: Please, shorten as much as you can the discussion.

Response 3: We have simplified the discussion section as much as possible.
